# Self-Reported Autistic Traits Using the AQ: A Comparison between Individuals with ASD, Psychosis, and Non-Clinical Controls

**DOI:** 10.3390/brainsci10050291

**Published:** 2020-05-14

**Authors:** Laura Fusar-Poli, Alessia Ciancio, Alberto Gabbiadini, Valeria Meo, Federica Patania, Alessandro Rodolico, Giulia Saitta, Lucia Vozza, Antonino Petralia, Maria Salvina Signorelli, Eugenio Aguglia

**Affiliations:** Department of Clinical and Experimental Medicine, Psychiatry Unit, University of Catania, Via Santa Sofia 78, 95123 Catania, Italy; laura.fusarpoli@gmail.com (L.F.-P.); alessia.ciancio@gmail.com (A.C.); alberto.gabbiadini@gmail.com (A.G.); valeriameo26291@gmail.com (V.M.); pataniafederica@gmail.com (F.P.); alessandro.rodolico@me.com (A.R.); giulia.saitta.91@gmail.com (G.S.); luciav85@gmail.com (L.V.); petralia@unict.it (A.P.); maria.signorelli@unict.it (M.S.S.)

**Keywords:** autism spectrum disorder, psychosis, schizophrenia, psychopathology, AQ, screening, accuracy, attention to detail, self-awareness, insight

## Abstract

The term “autism” was originally coined by Eugen Bleuler to describe one of the core symptoms of schizophrenia. Even if autism spectrum disorder (ASD) and schizophrenia spectrum disorders (SSD) are now considered two distinct conditions, they share some clinical features. The present study aimed to investigate self-reported autistic traits in individuals with ASD, SSD, and non-clinical controls (NCC), using the Autism-Spectrum Quotient (AQ), a 50-item questionnaire. The study was conducted in the Psychiatry Unit of Policlinico “G. Rodolico”, Catania, Italy. The AQ was administered to 35 adults with ASD, 64 with SSD, and 198 NCC. Overall, our data showed that the ASD sample scored significantly higher than NCC. However, no significant differences were detected between individuals with ASD and SSD. Notably, the three groups scored similarly in the subscale “attention to detail”. AQ showed good accuracy in differentiating ASD from NCC (AUC = 0.84), while discriminant ability was poor in the clinical sample (AUC = 0.63). Finally, AQ did not correlate with clinician-rated ADOS-2 scores in the ASD sample. Our study confirms that symptoms are partially overlapping in adults with ASD and psychosis. Moreover, they raise concerns regarding the usefulness of AQ as a screening tool in clinical populations.

## 1. Introduction

The term “autism” was firstly introduced by Eugen Bleuler (1911) to describe one of the core features of schizophrenia. Bleuler described autism as a “loss of contact with reality together with the relative and absolute predominance of the inner life” [1]. During the last century, several connotations were given to the term, until Leo Kanner (1943) described the neurodevelopmental disorder that is now called “autism” [2]. Only during the 1970s, autism and schizophrenia were regarded as very distinct entities [3,4]. Nowadays, autism spectrum disorder (ASD) is diagnosed in the presence of a persistent impairment in communication and reciprocal social interaction as well as restricted, repetitive patterns of behavior, interests, or activities. These symptoms usually occur during early childhood and cause significant impairments in everyday functioning. However, a diagnosis of ASD may be received later in life, when “social demands exceed the limited capacities of individuals” [5,6]. Prevalence estimates of ASD would range around 1.5% of the general population [7,8].

Schizophrenia spectrum disorders (SSD), also referred to as psychotic disorders, include instead a broad range of conditions which onset usually occurs during adolescence or young adulthood. They comprise not only schizophrenia, but also delusional disorder, brief psychotic disorder, schizophreniform disorder, schizoaffective disorder, psychoses induced by drugs or medical conditions, and psychoses not-otherwise-specified. SSD is characterized by heterogeneous symptoms which may vary in intensity and duration, such as hallucinations, delusions, disorganized speech, bizarre behaviors, and social withdrawal [5]. It has been estimated that approximately 1 in 150 individuals is diagnosed with a psychotic disorder at some point during their lifetime [9].

Even if SSD and ASD are currently considered distinct entities, they both represent chronic, multi-factorial disorders. They share genetic predispositions [10,11] and environmental risk factors, such as complications during pregnancy or at birth [12,13]. Moreover, they present with similar neuroimaging patterns [14,15], neurochemical abnormalities, such as dopaminergic dysregulations [16,17], and inflammatory pathways [18]. 

Both ASD and SSD show disturbances in several psychopathological domains; these alterations are similar in some cases, opposite in others [19]. First, content-thought disturbances may present with delusions in psychotic people, while scarce cognitive flexibility is typical of individuals with ASD. Paranoia is common to both conditions. However, in ASD it appears as a direct consequence of social interaction difficulties, rather than a cause of them. In fact, it has been hypothesized that the rigid thinking style may lead to difficulties in social interaction and thus to experience adverse social relations. Such events may produce negative beliefs which in turn lead to the onset of paranoid ideas [20]. Formal thought disturbances, such as the use of atypical or nonsensical language, characterized by tangentiality, circumstantiality, neologisms, are common to both groups [21].

Difficulties in social interaction are pervasive in both conditions and social cognitive deficits could partially explain the difficulties encountered by individuals with ASD and SSD [22]. However, other factors, such as the lack of interest in activities, the flat emotional affect, as well as the presence of thought disturbances (e.g., delusions), may play a critical role in psychoses [23]. A phenomenological analysis of the world–self boundary could help in an accurate differentiation: in fact, people with psychosis have a weak or variable boundary between the self and the world [24], while this limit seems better defined in individuals with ASD [25]. 

Perceptual alterations manifest in very different ways. Visual or auditory hallucinations are common in patients with psychoses [5], while hypo- or hyper-sensitivity is typical of individuals with ASD (e.g., the attraction for light sources, refusal of foods because of their color, elevated pain tolerance, altered olfactory threshold, etc.) [26,27]. Not by chance sensory alterations have been introduced among ASD core features in DSM-5 [5]. Behaviors might be disorganized in people with psychoses, while individuals with ASD typically feel comfortable with routines and sameness [28]. Nevertheless, mannerisms and stereotypies can occur in schizophrenia as well as in ASD [29]. Again, it is important to underline that the etiology is different. In SSD, mannerisms can emerge from delusional ideas, but may also be regarded as an expression of catatonic motor disorder or a manifestation of negativism [29]. The role of repetitive behaviors and mannerisms in individuals with ASD remains still unclear, although a wide variety of functions have been attributed to them: for instance, they can be used to calm anxiety, to communicate emotions, or for self-stimulatory purposes [30]. Interestingly, in the DSM-5 the specifier “with catatonia” has been introduced for ASD [5].

Even if ASD and SSD are currently considered two clearly distinct disorders, misdiagnoses are not infrequent, as clinicians who are not familiar with ASD may be misled by some peculiar features. For instance, the lack of meaningful relationships might be interpreted as an expression of negative symptoms (SSD) rather than a real difficulty in social interaction (ASD). Analogously, paranoia could be misjudged as an actual delusion (SSD) rather than a consequence of the difficulties in social cognition and theory of mind (ASD). Interestingly, Geurts et al. [31] reported that 9% of adults who received an ASD diagnosis in adulthood had been previously diagnosed with psychosis; this proportion was much higher (26%) in a study conducted by Nylander and Gillberg [32]. The co-occurrence of the two conditions represents another critical issue, as autistic symptomatology may be partially covered by a comorbid psychosis. In fact, on one hand, recent meta-analyses have reported that the pooled prevalence of SSD in individuals with ASD would range around 4% [33], 6% [34] or 9.5% [35]. One the other hand, the prevalence rates for autistic-like traits would range from 9.6% to 61% in psychotic patients, whilst the prevalence rates for diagnosed ASD ranged from <1% to 52% [36].

The numerous overlapping features between ASD and SSD may explain why people in the psychotic spectrum may misleadingly score above the cut-off in standardized diagnostic tools for ASD, such as the Autism Diagnostic Observation Schedule (ADOS-2) [37], as reported by several studies [6,38,39]. However, formal clinical evaluation for ASD with standardized tools is a long and time-consuming process [40]. Therefore, clinicians and researchers have tried to examine whether self-report instruments, such as the Autism-Spectrum Quotient (AQ) [41], could be useful in screening subjects with suspected ASD to address them to a more exhaustive evaluation. The AQ is a 50-item self-report tool that has been originally developed to measure the degree of autistic traits in adults with normal intelligence, with higher scores indicating more severe symptoms [41]. The AQ can be used for measuring autistic traits in the general population and for clinical screening of individuals with suspected ASD, with different cut-offs [42]. The guidelines of the UK National Institute for Health and Care Excellence (NICE) [40] suggest the use of the AQ-10—a brief version of the AQ [43]—as a screening tool for adults with possible autism. Moreover, the Adult Asperger Assessment (AAA), including the AQ, is suggested as a formal assessment tool to support the diagnosis of ASD in adults with intelligence within the normal range. Indeed, the AQ is used as a screening tool in clinical settings [44,45], as well as for the inclusion of participants in observational and interventional studies [46,47]. Interestingly, a large naturalistic study conducted by Ashwood et al. [48] has recently shown that self-reported AQ scores did not significantly predict receipt of a diagnosis of ASD in adulthood.

Focusing on the differences in AQ scores between ASD and SSD, a recent meta-analysis has found that people with SSD have indeed significantly higher autistic traits than the general population and lower autistic symptoms than individuals with ASD [49]. However, other authors have reported that, even if AQ may represent a reliable screening tool in the general population, its usefulness in identifying ASD in clinical environments is questionable [50,51,52]. Importantly, to our knowledge, only four papers specifically compared autistic traits in ASD and SSD and evaluated the discriminant ability of AQ between the two conditions, with contrasting findings [53,54,55,56]. In light of the inconsistent results regarding the usefulness and accuracy of AQ as screening tool among the general population as well as in psychiatric environments, the present study aimed to:Investigate the differences in self-reported autistic traits between adults with ASD, SSD and a non-clinical control group (NCC) from the general population;Analyze the accuracy of AQ in discriminating between ASD and SSD, as well as between ASD and NCC.Correlate the AQ scores with ADOS-2 scores in the ASD population.

## 2. Materials and Methods

### 2.1. Setting and Procedures

The present study was conducted in the outpatient service of the Psychiatry Unit of Policlinico “G. Rodolico”, Catania, Italy. From January to December 2019, we consecutively recruited 297 participants. Subjects were asked to complete a form containing personal information and to fill out the AQ. Each participant provided written informed consent before any study procedures commenced. The study was performed according to the Declaration of Helsinki and approved by our internal review board before recruitment.

### 2.2. Participants

The total sample comprised of 297 participants. For inclusion in the present study, all participants had to fulfill the following criteria: (1) age ≥ 18 years; (2) absence of intellectual disability or major cognitive impairment; (3) good knowledge of written and spoken Italian language; (4) written informed consent.

Thirty-five subjects had a diagnosis of ASD as confirmed by an exhaustive clinical examination and administration of standardized clinical interviews (i.e., Autism Diagnostic Observation Schedule-2 (ADOS-2) and/or Autism Diagnostic Interview-Revised (ADI-R); see a previous work by Fusar-Poli et al. [6] for detailed procedures). The ADOS-2 is a semi-structured observation of individuals who may belong to the autism spectrum. It is composed of different domains: communication, reciprocal social interaction, communication + social interaction, imagination/creativity, and stereotyped behaviors and restricted interests. The ADOS-2 consists of five modules addressed to children and adults according to their developmental and language levels. All participants included in the present study have been administered Module 4, which has been developed for adolescents and adults with good verbal fluency. For score calculation, we used the original algorithm proposed by Lord et al. [37]. According to the original algorithm, the domain “communication + social interaction” should be used to collocate an individual into the autism spectrum or autism. Of note, the presence of current or past psychiatric comorbid disorders was considered a reason for exclusion from the analysis.

Sixty-four subjects had received a diagnosis of SSD, as confirmed by a clinical evaluation made by at least two medical doctors (one senior psychiatrist and a trainee), and the administration of the Structured Clinical Interview for DSM-5 (SCID-5) [57]. Participants received the following diagnoses: unspecified psychosis (*n* = 22), paranoid schizophrenia (*n* = 9), schizoaffective disorder (*n* = 9), substance-induced psychosis (*n* = 9), delusional disorder (*n* = 5), unspecified schizophrenia (*n* = 5), undifferentiated schizophrenia (*n* = 2), catatonic schizophrenia (*n* = 1), residual schizophrenia (*n* = 1), disorganized schizophrenia (*n* = 1). None of the subjects were in a florid psychotic state at the moment of study completion, i.e., they did not present severe positive symptoms, profound negative symptoms, significantly disorganized or catatonic behaviors. The presence of ASD was excluded by a clinician with significant expertise in the field after the consultation of patients’ history through clinical charts and the direct observation of the subjects.

Finally, we recruited 198 non-clinical controls (NCC) among students and faculty staff members. Participants from the general population were interviewed by a senior psychiatrist using the SCID-5 [57]. People who fulfill the criteria for any psychiatric diagnosis were excluded from the analysis. The socio-demographic characteristics of the participants are reported in Table 1.

### 2.3. Autism-Spectrum Quotient (AQ)

All participants completed the AQ, the adult version, a widely used measure for the identification of autistic traits in the general population. Literature has shown that the reliability and consistency of the AQ are good [42]. The AQ consists of 50 items, rated using a 4-point Likert scale (1 = “definitely agree”, 2 = “slightly agree”, 3 = “slightly disagree”, and 4 = “definitely disagree”). It is composed of five subscales: social skills (SS), communication (C), imagination (I), attention to detail (AD), and attention switching (AS). We used the binary scoring method (the presence of autistic traits, either mildly or strongly, is scored as a +1, while the opposite is scored 0). Using the binary score method, the total score ranges can between 0 and 50, while the score of each subscale can range between 0 and 10. Higher AQ total score indicates higher autistic traits; higher scores in each subscale reflect poor social skills, poor communication skills, poor imagination, strong attention to details, and poor attention switching, respectively.

### 2.4. Statistical Analysis

Data were tested for normal distribution before applying statistical procedures. Continuous variables were reported as means and standard deviations, while dichotomous variables as percentages or counts, as appropriate. Chi-squared tests and one-way ANOVA were used to detect differences in socio-demographic characteristics between participants in the ASD, SSD, and NCC groups. One-way ANOVA was used also to investigate differences in AQ scores between the three groups. For post hoc between-group comparisons, the Tukey HSD test was applied.

Receiver operating characteristic (ROC) analyses were used to evaluate the accuracy of AQ in discriminating ASD from SSD and from NCC. We used the classification proposed by Hosmer et al. [58] for the interpretation of AUC values (0.5 = no discrimination; 0.51–0.69 = poor; 0.7–0.79: acceptable; 0.8–0.89: excellent; ≥0.9 = outstanding). Cohen’s k was used to calculate the agreement between clinical diagnosis and classification with ASDASQ. For data interpretation, we used the cutoffs proposed by Landis and Koch [59] (0 = no agreement; 0–0.2 = slight; 0.21–0.40 = fair; 0.41–0.60 = moderate; 0.61–0.80 = substantial; 0.81–1 = almost perfect agreement).

Results were considered statistically significant at the *p* ≤ 0.05 level, and all tests were two-tailed. Statistical analysis was performed using SPSS v. 23.0 software packages (IBM, Armonk, NY, USA).

## 3. Results

### 3.1. Characteristics of the Sample

We recruited a total of 297 subjects, of which 35 had a diagnosis of ASD, 64 had an SSD, and 198 did not meet the criteria for any psychiatric disorder. The sample was mainly composed of males (*n* = 157), who represented 52.8% of the sample, with no differences between the three groups. Participants were meanly 34.18 ± 18.57 years old (range 18–77), with the ASD group being younger than the SSD and the NCC groups. Significant differences were found also at the educational level, occupational and marital status. In fact, while ASD and SSD patients had completed mainly secondary or high school, a considerable part of controls had a university degree. Moreover, NCC were mostly employed; conversely, a large proportion of participants with ASD and SSD were unemployed, and 34.3% of individuals with ASD were students. Most participants were single, even if in NCC and SSD groups many subjects were married. The characteristics of participants and the ADOS-2 scores for the ASD group have been reported in Table 1.

### 3.2. Differences in AQ Scores

Overall, our sample (*n* = 297) obtained a mean score of 18.60 ± 7.88 at the AQ (range 3–43). The highest scores were obtained in the AS domain (4.65 ± 2.33) and the AD (4.55 ± 2.20) domains. A mean value of 3.32 ± 2.10 was scored in the I subscale, while the SS and C had overall mean scores of 3.09 ± 2.49 and 3.00 ± 2.40, respectively. The distribution of scores in the three groups is depicted in Figure 1.

One-way ANOVA detected significant differences between the three groups (*p* < 0.001) except for the AD domain, where no differences were found. However, Tukey HSD post-hoc analysis revealed that while both ASD and SSD significantly differed from NCC in all domains (excluding Imagination), no significant differences could be found between ASD and SSD patients, neither in the overall AQ score or subscales. The mean and SD for each group and the results of the statistical comparisons have been reported in Table 2.

### 3.3. Analysis of Accuracy

ROC curves showed that AQ had an excellent accuracy in differentiating individuals with ASD from NCC (AUC = 0.84, CI 95% 0.76–0.92, *p* < 0.001). On the contrary, the accuracy of AQ in discriminating individuals with ASD from SSD was poor (AUC = 0.63, 95% CI 0.51–0.75, *p* = 0.03). ROC curves are reported in Figure 2.

Table 3. reports the values of AUC, sensitivity, specificity, positive (PPV) and negative predictive values (NPV), and agreement with the diagnostic category. Notably, the agreement with the clinical group was fair in the case of NCC (k = 0.45) and null in the case of SSD (k = 0.04). For calculation, we considered a cut-off of ≥26 for the NCC group and ≥32 for the SSD group, as proposed by Ruzich et al. [42].

### 3.4. Correlation between AQ and ADOS-2 Scores

We computed Pearson’s correlation coefficients (r) to evaluate the correlation between AQ and ADOS-2. Substantially, we did not find any significant correlation, except for those between the AQ Imagination subscale and the social interaction, communication + social interaction and imagination domains of ADOS-2. The correlation matrix has been reported in Table 4.

## 4. Discussion

Our study examined the differences in AQ scores between individuals in the autism spectrum, in the schizophrenia spectrum and individuals from the general population, as well as the accuracy of the AQ in discriminating between the different groups. Our data showed that while AQ may represent a good instrument to detect autistic features among the general population (AUC = 0.84), it is not able to correctly discriminate between ASD and SSD (AUC = 0.63), with no significant differences either in the total score or in single subscales. Our results are in contrast with a recent meta-analysis [49] which found that patients in the SSD had lower autistic traits than ASD, but similar to the findings of Lugnegård et al. [55], who reported no significant differences in self-reported AQ scores between autistic and psychotic patients while using the full AQ scales, and poor discriminant validity of the questionnaire (AUC = 0.65).

The more reasonable explanation of our results is that ASD and SSD features are partially overlapping. In fact, the AQ evaluates areas which are typically impaired in both conditions, such as deficits in socio-communication, attention, and imagination. As mentioned above, abnormalities in verbal and non-verbal communication as well as in social cognition are common to both ASD and SSD. Attention switching, that is the capacity of an individual to flexibly shift mental set to different cognitive demands, is impaired in people with ASD, probably because the restriction of interests hampers them to switch between multiple clues [60,61]. Individuals affected by SSD show analogous impairment in switching attention, even if researchers have not yet clarified whether they should be ascribed to a primary deficit of attention or should be considered secondary to the emergence of delusions, or the experience of hallucinations [62]. Imagination represents instead “the faculty or action of forming new ideas, or images or concepts of external objects not present to the senses, typically derived from creative integration of past experiences, learning, or other information” [63]. Imagination is thought to be limited in individuals with autism, while over-developed in schizophrenia [64]. One can think about the “fantasy life” which characterized Bleuler’s autism [1]. However, as suggested by Spek and Wouters [65], most items of the AQ imagination subscale refer to active and purposeful imagination, i.e., “I find it difficult to imagine what it would be like to be someone else”. Despite the over-developed imagination in schizophrenia, active control in this respect has been found limited [66], and this could explain while this scale is not able to differentiate ASD from SSD.

Interestingly, in the ASD sample, the AQ scores of the scale regarding “attention to detail” (AD) did not significantly differ from SSD neither from the non-clinical group. Our finding is conflicting with the previous work by Lugnegård et al. [55], which instead found that ASD scored significantly higher in the AD domain than SSD and NCC. While they hypothesized that this subscale may comprise more ASD-specific items, we could not confirm this assumption, as our ASD sample scored similarly to the other groups. One potential explanation is that Lugnegård et al. have recruited subjects with DSM-IV Asperger’s syndrome, while our sample was composed of people with a DSM-5 diagnosis of ASD, thus including individuals with higher symptoms severity, even in presence of an IQ in the average range (the presence of ID was an exclusion criterion). Another explanation could be related to the different sex distribution, since in Lugnegård et al. the ASD sample comprised mainly women (51.9%), while our sample was predominantly composed of men (62.9%). However, this is just a speculation, and it is worth underlying that other authors did not find significant differences between ASD and SSD in the AD domain [65].

Another potential reason for our global findings is that the use of a self-report questionnaire, such as the AQ, may not be reliable in clinical contexts. It has in fact been reported that psychiatric patients—above all people in the schizophrenia spectrum—frequently present low levels of insight and tend to under- or over-report their symptoms [67,68]. Lack of self-awareness has been reported also in the population with autism, especially in the presence of greater functional impairment [69]. In fact, the use of self-report measures in the ASD population—including the AQ—has been questioned [70,71]. This hypothesis is partially confirmed by the low sensitivity shown by the AQ, which means a high rate of false negatives. In fact, according to our data, sensitivity was 22.9%, meaning that 77.1% of the ASD sample did not score above the cut-off suggested for clinical samples (≥32). Sensitivity improved (57.1%) while examining the accuracy of AQ un the general population, using the proposed cut-off of ≥26. This result sheds light on a significant limitation of the AQ, since a high sensitivity is clearly important for a screening tool. Nevertheless, it is worth underlying that the AQ has been developed as a descriptive, rather than a diagnostic measure of autistic traits, and for screening purposes rather than for differential diagnosis [41,42].

The poor insight of ASD participants may also explain why the AQ scores in our sample did not correlate with ADOS-2 scores. The ADOS-2 consists of a semi-structured observation of the individual’s behaviors and is rated by trained clinicians, not a self-reported tool. This finding is consistent with previous studies [48,52,72] which found no significant correlations between AQ total and ADOS-2 Module 4 scores. Conversely, it has been reported that AQ scores show reliable correlations with measures of anxiety, depression and alexithymia, suggesting that this instrument may be sensitive to non-specific mental-health vulnerabilities rather than to the defining characteristics of ASD specifically [73].

Despite the importance of our findings, several limitations should be highlighted. First, the sample size, especially the ASD group, was quite small; nevertheless, we have planned to enlarge our sample in future studies to replicate or disconfirm our findings. Second, the ASD group was younger than the SSD and NCC groups as we could match for sex, but not for age. Moreover, given the limited number of participants, we could not perform separate analyses based on sex. Some authors have argued the existence of a “female autistic phenotype”, according to which females in the autism spectrum may present with peculiar features, different from their male peers [74]. It would be interesting in future research to evaluate if screening tools, such as the AQ, work better with men or women. Third, we did not conduct a naturalistic study evaluating the predictive value of AQ for a subsequent diagnosis of ASD, as in Aswhood et al., for instance [48]. AQ questionnaires were administered only to individuals with ASD or psychoses, while no other psychiatric disorders were considered. For instance, obsessive-compulsive disorder or personality disorders present overlapping features with ASD, and the examination of AQ accuracy in these groups of patients would be equally useful. Finally, our study was conducted in a single Psychiatry Unit in Italy, therefore we cannot assure cross-cultural generalizability of our results.

## 5. Conclusions

Our study confirmed that the AQ may be useful in discriminating individuals with ASD from non-clinical controls. Nevertheless, it should be cautiously used for ASD screening in clinical populations, especially in the presence of psychotic patients. As suggested by other authors, AQ alone should not be used to exclude further ASD assessment other than if the scores are extremely low [52]. Therefore, the adoption of the AQ as a clinical tool (as recommended by NICE Guidelines [40]) may need to be reconsidered and adapted to different populations [48]. Future studies should investigate the intriguing relationship between insight and self-reported autistic traits. Furthermore, it would be interesting to evaluate the relationship between self-reported and clinician-rated measures in adults with ASD.

## Figures and Tables

**Figure 1 brainsci-10-00291-f001:**
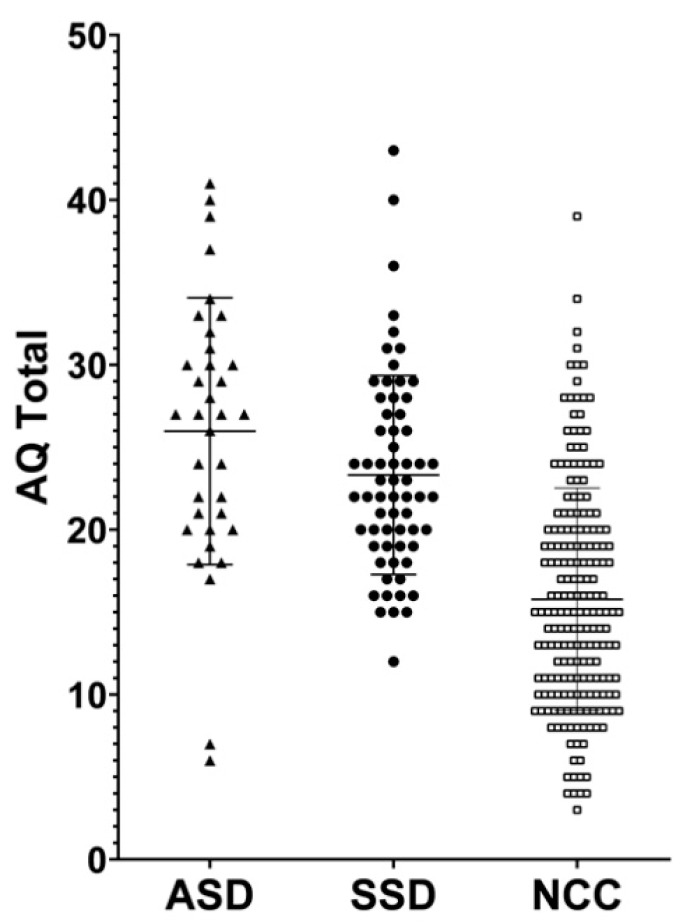
Distribution of AutismSpectrum Quotient (AQ) total scores among individuals with autism spectrum disorder (ASD), schizophrenia spectrum disorders (SSD) and non-clinical controls (NCC).

**Figure 2 brainsci-10-00291-f002:**
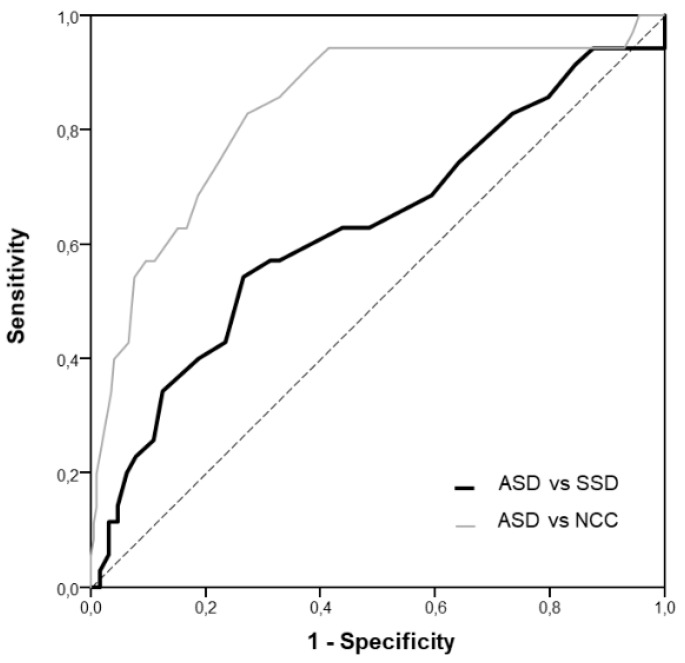
Receiver operating characteristic (ROC) curves of AQ total score.

**Table 1 brainsci-10-00291-t001:** Characteristics of participants.

	ASD Group (*n* = 35)	SSD Group (*n* = 64)	NCC (*n* = 198)	*p*-Value
Sex, male (%)	22 (62.9)	39 (60.9)	96 (48.5)	0.1
Age, mean ± SD (range)	26.15 ± 6.55 (18–45)	39.10 ± 14.48 (18–77)	34.01 ± 11.99 (19–67)	<0.001 *
Educational level, *n* (%)				<0.001 *
Primary school	0 (0)	8 (12.5)	0 (0)	
Secondary school	12 (34.3)	27 (42.2)	6 (3)	
High school	18 (5.4)	23 (35.9)	24 (12.1)	
University	5 (14.3)	6 (9.4)	168 (84.8)	
Occupational status, *n* (%)				<0.001 *
Full-time	5 (14.3)	9 (14.1)	114 (57.6)	
Part-time	4 (11.4)	0 (0)	14 (7.1)	
Unemployed	14 (40)	41 (64.1)	11 (5.6)	
Student	12 (34.3)	8 (12.5)	54 (27.3)	
Retired	0 (0)	6 (9.4)	5 (2.5)	
Marital status, *n* (%)				0.004 *
Single	33 (94.3)	43 (67.2)	129 (65.2)	
In a domestic partnership	1 (2.9)	2 (3.1)	22 (11.1)	
Married	1 (2.9)	11 (17.2)	38 (19.2)	
Divorced	0 (0)	7 (10.9)	6 (3)	
Widowed	0 (0)	1 (1.6)	3 (1.5)	
ADOS-2, mean ± SD				
Communication	3.62 ± 1.59 (0–6)	-	-	-
Social Interaction	6.74 ± 2.94 (2–16)	-	-	-
Communication + Social Interaction	10.4 ± 4.23 (2–22)	-	-	-
Imagination/Creativity	0.86 ± 0.65 (0–2)	-	-	-
Restricted Interests and Repetitive Behaviors	1.80 ± 1.30 (0–5)	-	-	-

ADOS-2 = Autism Diagnostic Observation Schedule-2; ASD = Autism Spectrum disorder; NCC = Non-clinical controls; SSD = Schizophrenia spectrum disorders. * Statistically significant.

**Table 2 brainsci-10-00291-t002:** AQ scores obtained by each group, and differences between groups.

				Overall		ASD vs. SSD	ASD vs. NCC	SSD vs. NCC
AQ Scores, Mean ± SD (range)	ASD (*n* = 35)	SSD (*n* = 64)	NCC (*n* = 198)	F	*p*	*p*	*p*	*p*
**AQ total**	25.97 ± 8.09 (6–41)	23.31 ± 6.03 (12–43)	15.77 ± 6.75 (3–39)	53.42	<0.001 *	0.15	<0.001 *	<0.001 *
**Social skills**	5.06 ± 2.55 (0–10)	4.02 ± 2.41 (0–10)	2.44 ± 2.23 (0–9)	25.73	<0.001 *	0.08	<0.001 *	<0.001 *
**Attention switching**	6.31 ± 2.42 (0–10)	5.66 ± 2.00 (1–10)	4.03 ± 2.15 (0–9)	25.59	<0.001 *	0.32	<0.001 *	<0.001 *
**Attention to detail**	4.97 ± 2.17 (1–9)	4.56 ± 2.22 (1–10)	4.47 ± 2.21 (0–10)	0.77	0.46	0.65	0.43	0.95
**Communication**	5.31 ± 2.31 (0–10)	4.48 ± 2.21 (1–10)	2.11 ± 1.93 (0–9)	58.63	<0.001 *	0.13	<0.001 *	<0.001 *
**Imagination**	4.31 ± 1.74 (1–7)	4.59 ± 1.81 (0–8)	2.73 ± 2.00 (0–10)	27.74	<0.001 *	0.77	<0.001 *	<0.001 *

AQ = Autism-spectrum quotient; ASD = Autism Spectrum disorder; NCC = Non-clinical controls; SSD = Schizophrenia spectrum disorders; * Statistically significant.

**Table 3 brainsci-10-00291-t003:** Accuracy of AQ in discriminating ASD from SSD and NCC.

	AUC (95% CI)	Sensitivity	Specificity	PPV	NPV	Cohen’s k
**ASD vs. SSD**	0.63 (0.51–0.75)	22.9%	92.2%	61.5%	68.6%	0.04
**ASD vs. NCC**	0.84 (0.76–0.92)	57.1%	90.4%	51.3%	92.3%	0.45

ASD = Autism spectrum disorder; AUC = Area Under Curve; CI = Confidence Interval; NCC = Non-clinical controls; NPV = Negative predictive value; PPV = Positive predictive value; SSD = Schizophrenia spectrum disorders.

**Table 4 brainsci-10-00291-t004:** Correlations between AQ and ADOS-2 scores.

		ADOS-2				
		Communication	Social Interaction	Communication + Social Interaction	Imagination/ Creativity	Repetitive Behaviors
**AQ**	**Total**	−0.09 *p* = 0.59	−0.01 *p* = 0.93	−0.03 *p* = 0.84	−0.09 *p* = 0.59	−0.02 *p* = 0.89
	**Social skills**	−0.32 *p* = 0.06	−0.24 *p* = 0.16	−0.29 *p* = 0.09	−0.28 *p* = 0.11	−0.22 *p* = 0.21
	**Attention switching**	−0.03 *p* = 0.87	−0.01 *p* = 0.99	−0.01 *p* = 0.96	−0.09 *p* = 0.59	0.1 *p* = 0.57
	**Attention to detail**	0.001 *p* = 0.99	−0.05 *p* = 0.76	−0.01 *p* = 0.98	−0.11 *p* = 0.54	0.03 *p* = 0.87
	**Communication**	0.11 *p* = 0.53	0.03 *p* = 0.87	−0.02 *p* = 0.91	−0.09 *p* = 0.59	−0.15 *p* = 0.40
	**Imagination**	0.21 *p* = 0.21	0.37 *p* = 0.03 *	0.35 *p* = 0.04 *	0.43 *p* = 0.01 *	0.29 *p* = 0.09

* Statistically significant correlations with *p* < 0.05.

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
