# Peer review of "Self-Reported Autistic Traits Using the AQ: A Comparison between Individuals with ASD, Psychosis, and Non-Clinical Controls"

_brainsci, 2020, doi:10.3390/brainsci10050291_

Round 1

Reviewer 1 Report

The ms. examined the applicability of AQ (self-reported questionnaire for autistic traits) to distinguish autism from healthy and clinical control group. The authors found that AQ may represent a good instrument to detect autistic features among the general population but has poor ability to discriminate autism from schizophrenia.

First of all, I think this ms. is a bit out of scope for the Brain Sciences as I did not find here anything related to neuroscience and brain. These results might be more appropriate for some clinical journal, related to autism or alike.

Secondly, I think the authors have to elaborate a bit on the problem they address. Are there many clinicians/researchers who are using AQ to differentiate between clinical diagnoses? As authors correctly mentioned AQ initially was not designed for this task. Other issue is the small size of clinical sample, as correctly pointed out in limitation section of the ms.

Author Response

Q1: The ms. examined the applicability of AQ (self-reported questionnaire for autistic traits) to distinguish autism from healthy and clinical control group. The authors found that AQ may represent a good instrument to detect autistic features among the general population but has poor ability to discriminate autism from schizophrenia.

First of all, I think this ms. is a bit out of scope for the Brain Sciences as I did not find here anything related to neuroscience and brain. These results might be more appropriate for some clinical journal, related to autism or alike.

R1: We thank the Reviewer for the comment. Actually, the manuscript has been submitted for the Special Issue “Advances in Autism Research” which collects papers with significant translational effect to the field of clinical services, including those related to the screening and diagnosis of ASD.

Q2: Secondly, I think the authors have to elaborate a bit on the problem they address. Are there many clinicians/researchers who are using AQ to differentiate between clinical diagnoses? As authors correctly mentioned AQ initially was not designed for this task.

R2: Thank you for the really meaningful comment. We have tried to better explain this issue in the Introduction and the Conclusions.

“The guidelines of the UK National Institute for Health and Care Excellence (NICE) [40] suggest the use of the AQ-10 - a brief version of the AQ [43] – as a screening tool for adults with possible autism. Moreover, the Adult Asperger Assessment (AAA), including the AQ, is suggested as a formal assessment tool to support the diagnosis of ASD in adults with intelligence within the normal range. Indeed, the AQ is used as screening tool in clinical settings [44, 45], as well as for inclusion of participants in observational and interventional studies [46, 47]. Interestingly, a large naturalistic study conducted by Ashwood et al. [48] have recently shown that self-reported AQ scores did not significantly predict receipt of a diagnosis of ASD in adulthood.

“Our study confirmed that the AQ may be useful in discriminating individuals with ASD from non-clinical controls. Nevertheless, it should be cautiously used for ASD screening in clinical populations, especially in the presence of psychotic patients. As suggested by other authors, AQ alone should not be used to exclude further ASD assessment other than if the scores are extremely low [52]. Therefore, the adoption of the AQ as a clinical tool (as recommended by NICE Guidelines [40]) may need to be reconsidered and adapted to different populations [48]. “

Q3: Other issue is the small size of clinical sample, as correctly pointed out in limitation section of the ms.

R3: We agree with the Reviewer. In fact, we have pointed out this important limitation – that cannot be modified at this point - in the Discussion. However, we would like to underline that our sample [N = 297] is larger than the samples of all previous similar studies (see Wouters and Spek 2011 [N = 63], Naito et al. 2010 [N = 97], Lugnegård et al. 2015 [N = 136], Zhang et al. 2016 [N = 145, including also OCD patients]).

Reviewer 2 Report

Review:

Self-reported autistic traits using the AQ: a comparison between patients with ASD, psychosis and non-clinical controls.

  1. English language

There are some grammar changes necessary. For instance, see lines: 38, 45, 48, 57, 68, 75, 124, etc.

Also, the term ‘ASD people’ is used as well as the term ‘autistic individuals’ or ‘people with ASD’. Maybe it is possible to use one term only throughout the hole paper. The term ‘individuals with autism’ is preferred.

  1. Instruments

All participants had to fulfill the criteria: absence of intellectual disability or major cognitive impairment. How did the researchers measure this?

I do not understand why the researchers used the SCID 5 interview for confirming the schizophrenia spectrum disorder. The SCID 5 is a personality disorder questionnaire or interview. It is not designed for detecting or classifying schizophrenia. Could this be explained?

  1. Overall

The researchers state that ASD and SSD share many clinical features. When looking at the DSM 5, SSD are defined by abnormalities in delusions, hallucinations, disorganized thinking, grossly disorganized or abnormal motor behavior (including catatonia), and negative symptoms. These abnormalities are not seen in ASD. The researchers state that deficits in social functioning are typical in both ASD and in SSD. However, they do not explain that these deficits have different sources. This is also the case for the mentioned mannerisms and motor stereotypies. The researchers do mention one phrase in the discussion section:

“Individuals affected by SSD show analogous impairment in switching attention, even if it is not clear whether they should be ascribed to a primary deficit of attention or should be considered secondary to the emergence of delusions, or the experience of hallucinations.” I would consider this as an insuperable limitation of the study.

Furthermore, the researchers state that ‘several authors’ (no references are mentioned) have examined the overlap and co-occurrence of the two conditions. However, the DSM 5 does not mention such comorbidity. The researchers state that there are ‘numerous overlapping features’. Maybe the researchers could limit the overlapping features to the subscales of the AQ (social skills, communication, imagination, attention to detail and attention switching)? These subscales represent 5 areas associated with autism.

The above comments could be added and explained in the paper.

Another challenging limitation of the study is the present low levels of insight and lack of self-awareness in ASD and SSD individuals. Maybe the results would be different when the AQ was administered to parents, relatives or other caregivers around the clients. Why was that not considered? Especially because the researchers already found that the AQ scores in the sample did not correlate with the ADOS-2 scores.

In the conclusion the researchers state that they confirmed that the AQ represent a useful tool for evaluation of autism traits in the general population. However, this was not a research question.

In general, I have significant concerns about the study. In the abstract the researchers state that their study aims to investigate self-reported autistic traits in individuals with ASD, SSD and non-clinical controls using the AQ (which, however, is not mentioned as a research question!!). Why are autistic traits in SSD examined? Why did the researchers only use the AQ and did not administer the EQ and SQ also? The use of the AQ, EQ and SQ combined, is highly recommended. Why did the researchers want to measure the accuracy of the AQ? This is not explained by the researchers.

Author Response

Q1: There are some grammar changes necessary. For instance, see lines: 38, 45, 48, 57, 68, 75, 124, etc.

R1: We thank the Reviewer for the comment. We have corrected the English language and grammar according to the Reviewer’s suggestions.

Q2: Also, the term ‘ASD people’ is used as well as the term ‘autistic individuals’ or ‘people with ASD’. Maybe it is possible to use one term only throughout the hole paper. The term ‘individuals with autism’ is preferred.

 R2: We thank the Reviewer for the comment. We have changed the terminology throughout the paper.

Q3: All participants had to fulfill the criteria: absence of intellectual disability or major cognitive impairment. How did the researchers measure this?

R3: The absence of intellectual disability or major cognitive impairment was ascertained using standardized instruments  (e.g. WAIS-IV, Raven’s Matrices, etc.) only for the ASD sample. In fact, we routinely administer IQ tests during the first extensive clinical evaluation. On the contrary, the absence of cognitive impairment in the SSD and NCC groups was based only on clinical judgment. Please note, that all participants had been in contact with our Unit for much time (>3 months) and thus were well-known by the clinicians involved in the study.

Q4: I do not understand why the researchers used the SCID 5 interview for confirming the schizophrenia spectrum disorder. The SCID 5 is a personality disorder questionnaire or interview. It is not designed for detecting or classifying schizophrenia. Could this be explained?

R4: As the Reviewer probably already knows, in DSM-IV-TR there was a distinction between Axis I and Axis II disorders (personality disorders), which could be respectively evaluated using the SCID-I and SCID-II. However, this distinction is not present in the DSM-5: in fact, the SCID-5 is a semi-structured interview divided in several modules which can be used as a guide for making all major DSM-5 diagnoses - including schizophrenia spectrum disorders - and not only personality disorders (see First 2015). Some studies in which the SCID-5 was used for diagnostic confirmation are for example Steardo, L. et al. 2020 https://doi.org/10.1186/s12991-020-00266-7 (Bipolar disorder); Freeman et al. 2020 https://doi.org/10.1002/da.23017 (Major depression); Bener et al 2018 https://doi.org/10.1002/da.23017 (Schizophrenia and OCD); Bonflis et al. 2019 https://doi.org/10.1093/schbul/sby137 (Schizophrenia and schizoaffective disorder).

Q5: The researchers state that ASD and SSD share many clinical features. When looking at the DSM 5, SSD are defined by abnormalities in delusions, hallucinations, disorganized thinking, grossly disorganized or abnormal motor behavior (including catatonia), and negative symptoms. These abnormalities are not seen in ASD.

R5: We thank the Reviewer for the meaningful comment. We are aware that not all clinical features are present in both diagnostic categories, particularly those mentioned by the Reviewer. However, our purpose was to emphasize that similar features might be misinterpreted by clinicians who are not familiar with ASD. We have clarified this point in the Introduction

“Even if ASD and SSD are currently considered two clearly distinct disorders, misdiagnoses are not infrequent, as clinicians who are not familiar with ASD may be misled by some peculiar features. For instance, the lack of meaningful relationships might be interpreted as an expression of negative symptoms (SSD) rather than a real difficulty in social interaction (ASD). Analogously, paranoia could be misjudged as an actual delusion (SSD) rather than a consequence of the difficulties in social cognition and theory of mind (ASD). Interestingly, Geurts et al. [31] reported that 9% of adults who received an ASD diagnosis in adulthood had been previously diagnosed with psychosis; this proportion was much higher (26%) in a study conducted by Nylander and Gillberg [32]. The co-occurrence of the two conditions represents another critical issue, as autistic symptomatology may be partially covered by a comorbid psychosis. In fact, on one hand, recent meta-analyses have reported that the pooled prevalence of SSD in individuals with ASD would range around 4% [33], 6% [34] or 9.5% [35]. One the other hand, the prevalence rates for autistic-like traits would range from 9.6% to 61% in psychotic patients, whilst the prevalence rates for diagnosed ASD ranged from < 1% to 52% [36].

Q6: The researchers state that deficits in social functioning are typical in both ASD and in SSD. However, they do not explain that these deficits have different sources. This is also the case for the mentioned mannerisms and motor stereotypies.

R6: We thank the Reviewer for the comment. We have now clarified these points

Difficulties in social interaction are pervasive in both conditions and social cognitive deficits could partially explain the difficulties encountered by individuals with ASD and SSD [22]. However, other factors, such as the lack of interest in activities, the flat emotional affect, as well as the presence of thought disturbances (e.g. delusions), may play a critical role in psychoses [23]. A phenomenological analysis of the world-self boundary could help in an accurate differentiation: in fact, people with psychosis have a weak or variable boundary between the self and the world [24], while this limit seems better defined in individuals with ASD [25]. “

Nevertheless, mannerisms and stereotypies can occur in schizophrenia as well as in ASD [29]. Again, it is important to underline that the etiology is different. In SSD, mannerisms can emerge from delusional ideas, but may also be regarded as an expression of catatonic motor disorder or a manifestation of negativism [29]. The role of repetitive behaviors and mannerisms in individuals with ASD remains still unclear, although a wide variety of functions have been attributed to them: for instance, they can be used to calm anxiety, to communicate emotions, or for self-stimulatory purposes [30]. Interestingly, in the DSM-5 the specifier “with catatonia” has been introduced for ASD [5].

Q7: The researchers do mention one phrase in the discussion section: “Individuals affected by SSD show analogous impairment in switching attention, even if it is not clear whether they should be ascribed to a primary deficit of attention or should be considered secondary to the emergence of delusions, or the experience of hallucinations.” I would consider this as an insuperable limitation of the study.

R7: Thank you for the comment. We apologize for not being clear. The sentence mentioned by the Reviewer has been used to describe one potential explanation of attentional impairments in individuals with schizophrenia and should not be generalized to our sample. In fact, none of the psychotic patients who have completed the questionnaire were in a florid phase of their pathology. In the presence of vivid hallucinations or major perceptual dysfunctions, it would be almost impossible for them to complete the questionnaires.

We have now rephrased the sentence in the Discussion for clarification:

Attention switching, that is the capacity of an individual to flexibly shift mental set to different cognitive demands, is impaired in people with ASD, probably because the restriction of interests hampers them to switch between multiple clues [60, 61]. Individuals affected by SSD show analogous impairment in switching attention, even if researchers have not yet clarified whether they should be ascribed to a primary deficit of attention or should be considered secondary to the emergence of delusions, or the experience of hallucinations [62].

Moreover, we have clarified this point in the Methods section

“None of the subjects were in a florid psychotic state at the moment of study completion, i.e. they did not present severe positive symptoms, profound negative symptoms, significantly disorganized or catatonic behaviors.”

Q8: Furthermore, the researchers state that ‘several authors’ (no references are mentioned) have examined the overlap and co-occurrence of the two conditions. However, the DSM 5 does not mention such comorbidity.

R8: We thank the Reviewer for the comment. It is true that comorbid diagnosis between ASD and schizophrenia was not possible according to older versions of the DSM. For instance, the DSM-III stated that “autism could not occur in the presence of delusions, hallucinations, loosening of associations, or incoherence”. However, this “trumping rule” has been eliminated by the DSM-5. In fact, among DSM-5 specifiers for ASD it is possible to find: 1) Associated with another neurodevelopmental, mental, or behavioral disorder; 2) Catatonia. Therefore, the DSM-5 acknowledges the possibility of a comorbid diagnosis. The amount of evidence on the topic is quite large, as indicated by the meta-analyses cited in the Introduction of the manuscript, regarding both the prevalence of SSD in individuals with ASD (Lai et al. 2019; Lugo-Marin et al. 2018; De Giorgi et al., 2019) and the prevalence of ASD and autistic-like traits in psychotic patients (Kincaid et al 2017). See also R5.

Q9: The researchers state that there are ‘numerous overlapping features’. Maybe the researchers could limit the overlapping features to the subscales of the AQ (social skills, communication, imagination, attention to detail and attention switching)? These subscales represent 5 areas associated with autism.  The above comments could be added and explained in the paper.

R9: We thank the Reviewer for the comment. Indeed, in the Introduction of the paper, we preferred to give a more comprehensive overview of the two conditions, describing the psychopathological alterations presented by the two conditions and focusing not only on overlapping, but also on divergent areas, and leading to the rationale of the study. Specifically, we have analyzed formal and content thought disturbances, difficulties in social interaction, perceptual alterations, behavioral and motor alterations. Moreover, we believe that this section has much improved thanks to the comments of the Reviewer. Contrarywise, in the Discussion of the paper, we have focused exclusively on the subscales of the AQ, trying to critically examine potential explanations for our findings. We believe that presenting different information regarding ASD and schizophrenia (and not simply repeat the same notions) might be relevant for the clinical implications of the paper.

Q10: Another challenging limitation of the study is the present low levels of insight and lack of self-awareness in ASD and SSD individuals. Maybe the results would be different when the AQ was administered to parents, relatives or other caregivers around the clients. Why was that not considered? Especially because the researchers already found that the AQ scores in the sample did not correlate with the ADOS-2 scores.

R10: We thank the Reviewer for the comment. The AQ-Adult version has been developed only as a self-report tool, in contrast with other versions of the AQ (e.g. the AQ-Child version), that should be completed by parents or guardians. Therefore, it would have been methodologically incorrect to administer the questionnaire to parents or other caregivers. Moreover, the lack of self-awareness is one of the main issues that we wanted to point out with our paper, i.e. even if patients with psychiatric disorders, such as ASD or schizophrenia, frequently lack of self-insight, self-report measures (e.g. AQ) are still frequently used in clinical practice (see also R2 to Reviewer 1). The discrepancies between self-report and parent- or clinician-reported measures in ASD may represent another intriguing line of research that we would like to investigate in depth in future studies, as stated at the end of the manuscript; however, they are out of the scope of the present paper.

Q11: In the conclusion the researchers state that they confirmed that the AQ represent a useful tool for evaluation of autism traits in the general population. However, this was not a research question.

R11: We thank the Reviewer for the comment. We have now rephrased the conclusion:

“Our study confirmed that the AQ may be useful in discriminating individuals with ASD from non-clinical controls.

Q12: In general, I have significant concerns about the study. In the abstract the researchers state that their study aims to investigate self-reported autistic traits in individuals with ASD, SSD and non-clinical controls using the AQ (which, however, is not mentioned as a research question!!). Why are autistic traits in SSD examined? Why did the researchers only use the AQ and did not administer the EQ and SQ also? The use of the AQ, EQ and SQ combined, is highly recommended.

R12: We thank the Reviewer for the comment. We did not administer the EQ nor SQ since these instruments are not used to evaluate autistic traits, but empathizing and systemizing abilities. It would be very interesting to put the three instruments together and see the differences between ASD, SSD, and NCC. However, the present study was primarily aimed at evaluating the capacity of AQ, a widely used screening measure, to discriminate people with ASD to other clinical and non-clinical populations. We have now clarified the research questions at the end of the Introduction.

“Focusing on the differences in AQ scores between ASD and SSD, a recent meta-analysis has found that people with SSD have indeed significantly higher autistic traits than the general population and lower autistic symptoms than individuals with ASD [49]. However, other authors have reported that, even if AQ may represent a reliable screening tool in the general population, its usefulness in identifying ASD in clinical environments is questionable [50-52]. Importantly, to our knowledge, only four papers specifically compared autistic traits in ASD and SSD and evaluated the discriminant ability of AQ between the two conditions, with contrasting findings [53-56]. In light of the inconsistent results regarding the usefulness and accuracy of AQ as screening tool among the general population as well as in psychiatric environments, the present study aimed to:

  1. Investigate the differences in self-reported autistic traits between adults with ASD, SSD and a non-clinical control group (NCC) from the general population;
  2. Analyze the accuracy of AQ in discriminating between ASD and SSD, as well as between ASD and NCC.
  3. Correlate the AQ scores with ADOS-2 scores in the ASD population. “

Q13. Why did the researchers want to measure the accuracy of the AQ? This is not explained by the researchers.

R13: We thank the Reviewer for the observation. The accuracy of AQ was measured to evaluate the discriminant ability of this tool in different populations. In fact, statistical procedures like ANOVA only allow the comparison of the mean scores of the groups. However, for instance, we could have a mean score of  2 in the first group, 6 in the second, and 17 in the third. Is the difference between groups statistically significant? Probably yes. But, in this case, the means of the three groups would be below the cut off (26), which is not helpful. Here comes into play the accuracy: the AUC allows calculate the main psychometric properties of the instruments (e.g. sensitivity, specificity, positive and negative predictive values) based on the number of true positive/negative and false positive/negative. Therefore, it is useful to predict whether an instrument is able to correctly classify a case or control. A good accuracy (and a good sensitivity, in particular) is critical for a screening tool. The methodology adopted in our study is analogous to previous similar studies. See for instance Lugnegård et al. 2015.

Round 2

Reviewer 1 Report

I'm satisfied with the authors' response.